# Structural basis for the activation and ligand recognition of the human oxytocin receptor

Yann Waltenspühl [1,2], Janosch Ehrenmann[1,3,4], Santiago Vacca [1,4], Cristian Thom [1,4], Ohad Medalia [1] & Andreas Plückthun [1✉]

The small cyclic neuropeptide hormone oxytocin (OT) and its cognate receptor play a central role in the regulation of social behaviour and sexual reproduction. Here we report the single-particle cryo-electron microscopy structure of the active oxytocin receptor (OTR) in complex with its cognate ligand oxytocin. Our structure provides high-resolution insights into the OT binding mode, the OTR activation mechanism as well as the subtype specificity within the oxytocin/vasopressin receptor family.

[1] Department of Biochemistry, University of Zürich, Winterthurerstrasse 190, CH-8057 Zürich, Switzerland. [2]Present address: Novo Nordisk A/S, Novo Nordisk Park 1, DK-2760 Måløv, Denmark. [3]Present address: leadXpro AG, PARK innovAARE, CH-5234 Villigen, Switzerland. [4]These authors contributed equally: Janosch Ehrenmann, Santiago Vacca, Cristian Thom. ✉email: plueckthun@bioc.uzh.ch

The neurohypophyseal hormones oxytocin (OT) and arginine vasopressin (AVP) are cyclic peptides that activate an evolutionary ancient subfamily of class A G protein-coupled receptors (GPCRs)[1], comprising the oxytocin receptor (OTR) and the closely related vasopressin receptors ($V_{1A}R$, $V_{1B}R$ and $V_2R$). This OT/AVP hormone system is highly conserved among many species and exerts a central role in the regulation of social cognition, social behaviour, and sexual reproduction[2].

Currently, several clinical trials are evaluating the efficacy of OTR-mediated signalling through administration of OT to treat malfunctions such as autism-spectrum disorders[3], anxiety[4] and schizophrenia[5]. While OT itself is an approved peripheral drug in obstetrics[6], OT-based treatments of socio-behavioural deficiencies, requiring central administration of the hormone, have yet failed, potentially due to its poor drug-like properties[7] and limited penetration through the blood brain barrier[8]. Despite recent advances in finding a non-peptide agonist that is active in an animal model[9], the high demand for such drugs continues. Up to now, the identification and development of OTR-specific molecules with satisfactory pharmacokinetic properties, favourable biodistribution and specificity has been impeded by the lack of structural information on the OTR:OT signalling complex.

Here, we now report the single-particle cryo-electron microscopy (cryo-EM) structure of the OT-bound human OTR in complex with a heterotrimeric G protein at a resolution of 3.2 Å.

## Results and discussion

**Receptor and G protein engineering**. Initially, structural studies of the OTR:OT signalling complex were hampered by the poor biophysical behaviour of the wild-type OTR (wtOTR). To improve expression and purification yields we included a single stabilising mutation (D153Y), which we previously identified by a next-generation sequencing (NGS)-based in-depth analysis of directed evolution experiments[10]. This mutation enabled a 50-fold increase in the yield of purified functional receptor, with very similar agonist binding and signalling behaviour (Supplementary Fig. 1a, b). As an additional hurdle, complexes of OTR with an otherwise frequently employed engineered mini-$G_s$[11,12] or a $G_{s/q}$ chimera were not stable and dissociated upon plunge freezing. While $G_q$-based signalling is the main route of OTR activation, the receptor has been shown to also interact with $G_o$ and $G_i$[13], but not with $G_s$. Therefore, we hypothesised that the observed instability of the OTR:mini-$G_{s/q}$ complex may be attributed to unfavourable interactions of OTR with the $G_s$ domain, and interactions of the $G_q$ α5 helix are not sufficient to overcome this. Thus, to maximally stabilise the OTR active state, we designed a G protein chimera using mini-$G_o$[11] as a basis, substituting the $G_o$ α5 helix with the corresponding amino acids of $G_q$. Additionally, we replaced the N terminus with the respective $G_{i1}$ residues to permit binding of the complex-stabilising single-chain variable fragment 16 (scFv16)[14]. For simplicity, the resulting mini-G protein is denoted $G_{o/q}$ henceforth. These combined engineering efforts finally enabled single-particle cryo-EM analysis of the OTR:OT:$G_{o/q}$:scFv16 complex at a resolution of 3.2 Å (Fig. 1a, b, Supplementary Figs. 1c, d, 2–4 and Supplementary Table 1), enabling unprecedented insights into OTR activation by OT and the receptor G protein interaction.

**OT binding mode**. In the orthosteric ligand binding pocket, all nine amino acids of OT participate in OTR binding (Fig. 1c). The cyclic part (residues 1–6) is buried deep inside the pocket while the C-terminal tripeptide (residues 7–9) is facing the extracellular loops. Interestingly, the amidated C terminus of Gly[9], known to be important for activation[15], is located in proximity of residues E42[1.35] and D100[2.65] (Fig. 1d; numbers in superscripts

correspond to Ballesteros-Weinstein numbering[16]) of transmembrane helices I and II, which are involved in magnesium-dependent modulation of OT binding[17]. Together with the neighbouring Leu[8], the extracellular surface of the orthosteric pocket is lined by Pro[7] and Asn[5], which pack against extracellular loop 3 (ECL3) and ECL2, respectively. Leu[8] is oriented towards the extracellular space, explaining why position 8 is best suited for the attachment of fluorophores in OT[18]. Gln[4] is the only residue pointing out perpendicularly from the ring plane and stabilizes the cyclic ring position through a hydrogen bond to Q295[6.55]. Ile[3] is buried in a hydrophobic pocket formed by side-chain residues of transmembrane helices IV, V, and VI. The critical contribution of this hydrophobic pocket is underlined by the observed reduction in OT potency when mutating the main contact residues I201[5.39], I204[5.42] and F291[6.51] of the receptor to alanine (Fig. 1d, e and Supplementary Table 2). Tyr[2] penetrates deep into a crevice at the bottom of the orthosteric pocket formed by residues from helices II, III, VI and VII. While the carboxy group of Tyr[2] interacts with Q171[4.60], the phenol ring engages in hydrophobic interactions with Q92[2.57] and F291[6.51], and the hydroxyl group forms a hydrogen bond to the backbone amide oxygen of L316[7.40]. The importance of these interactions explains the loss of potency when either Q171[4.60] or F291[6.51] are mutated to alanine (Fig. 1d, e). Finally, Cys[1], which stabilizes the OT ring conformation through a disulphide with Cys[6], contacts with its backbone oxygen a polar cluster consisting of residues Q96[2.61], K116[3.29], and Q119[3.32] observed in the OT/AVP family, consistent with earlier mutagenesis studies[19] (Fig. 1d, e).

**OT mediated receptor activation**. A comparison of the OTR active-state structure with the previously reported inactive-state structure of the OTR in complex with the small-molecule antagonist retosiban[17] enabled us to identify the molecular changes involved in receptor activation (Figs. 2 and 3). In contrast to OT, retosiban only partially occupies the region of the orthosteric pocket, where the cyclic part of OT binds (Fig. 2b, c). Nonetheless, the OT-induced helical rearrangements in the orthosteric pocket are relatively small, reflected by the subtle change of the pocket volume between the active and inactive state (Fig. 2d, e). OT interacts with residues F291[6.51] and F292[6.52] at the bottom of the binding pocket. This interaction induces a rearrangement of F291[6.51], initiating the large outward movement of helix VI through a series of side-chain reorientations in conserved microswitches, including W288[6.48] (CWxP motif), F284[6.44] (PIF motif) and Y329[7.53] (NPxxY motif) (Fig. 3c, d). These rearrangements ultimately lead to the breakage of the interaction between T273[6.33] and R137[3.50] of the DRY motif (DRC in OTR), and the reorientation of D136[3.49] into a position enabling direct contact with the α5 helix of $G_{o/q}$ (Fig. 3e).

Early functional studies on OT derivatives identified Cys[1], Tyr[2] and Gln[4] as centrally involved ligand residues in receptor activation. For example, alkylation of the Tyr[2] hydroxy group led to decreased agonistic activity[20], suggesting an important role of this residue, consistent with our structural data. Remarkably, we observe a local unfolding of helix VII at the extracellular receptor side in the region of L316[7.40], creating a pronounced kink which is stabilised by a hydrogen bond formed between Tyr[2] of OT with the backbone oxygen of L316[7.40] (Fig. 3b). Importantly, a similar helix VII conformation was also observed in active-state structures of $V_2R$[21] (Supplementary Fig. 5), suggesting that partial helix VII unfolding is a feature of the OT/AVP family receptor activation. Sequence alignments of the four receptors of the OT/AVP family reveal that all receptors share a conserved kink region, with the exception of position 7.42, where both OTR and $V_2R$ share an alanine, whereas $V_{1A}R$ and

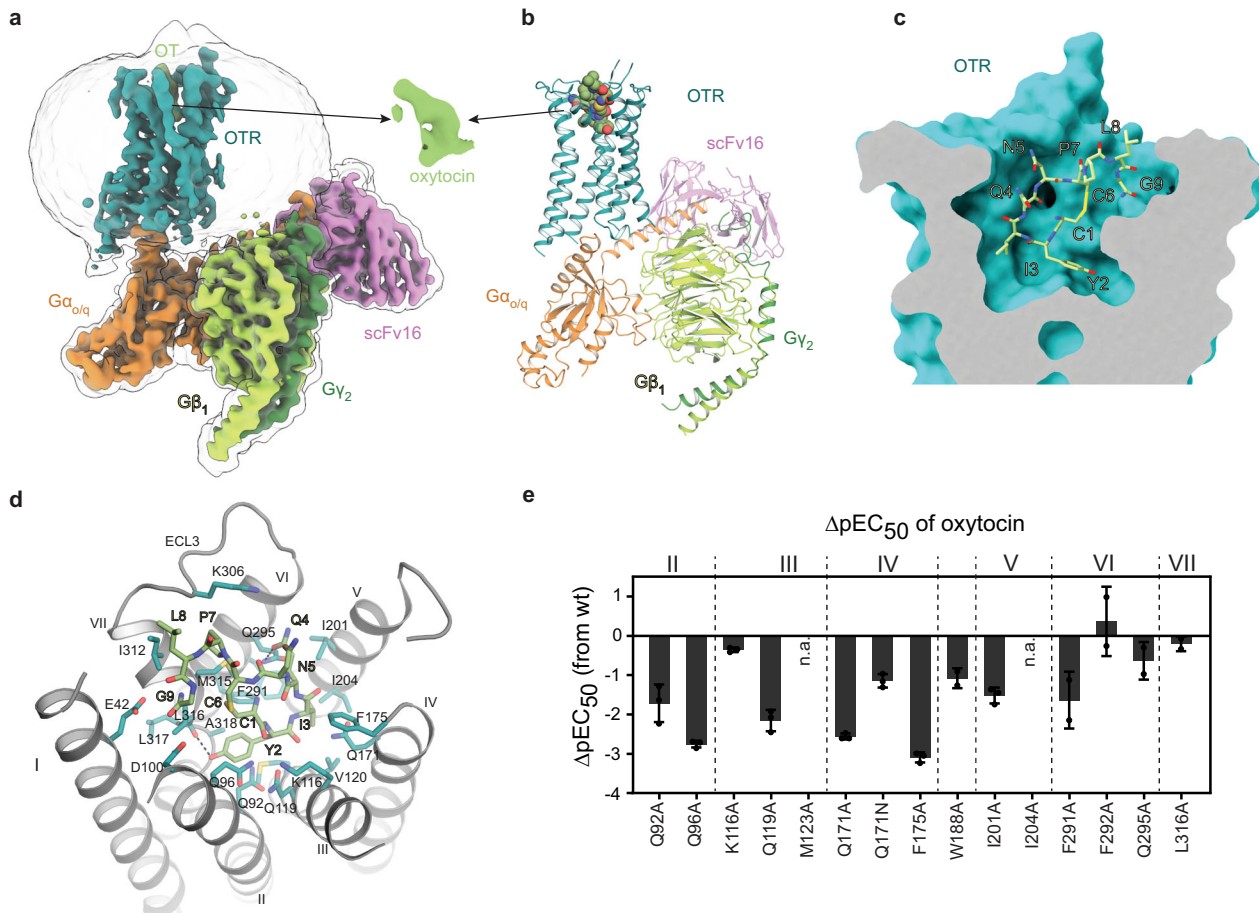

**Fig. 1 Overall structure of the oxytocin bound OTR. a** Cryo-EM map of the OTR:OT:$G_{o/q}$:scFv16 complex. **b** Molecular model of the OTR:OT:$G_{o/q}$:scFv16 complex. Ligand atoms are coloured according to atom types. **c** Clipped surface representation of the OT orthosteric binding pocket with OT shown as bright green sticks. **d** Detailed interactions of OT with OTR. **e** Agonist potency profiles of selected mutants. Bars represent differences in IP1 accumulation potency (mean $\Delta pEC_{50}$ ± standard deviation from two (W188A, F291A, F292A, Q295A, L316A) or three (all other constructs) independent experiments in duplicates) compared to wtOTR. n.a., no activation. Data are provided in Supplementary Table 2 and as a Source Data file.

$V_{1B}R$ carry a glycine. To test if a kink region carrying a glycine is compatible with the observed helix VII reorientation, we determined ligand binding affinity and receptor activation of an OTR mutant where we mutated A318[7.42] to glycine. We observe only little differences in activity and binding affinity, supporting an unaltered activation mechanism. It appears that glycine might potentially destabilise the local helical conformation and even facilitate ligand binding, as indicated by the slightly improved affinity and potency of OT to A318[7.42]G. Accordingly, we find that removal of the glycine in $V_{1A}R$ by mutation G337[7.42]A has the inverse effect (Supplementary Fig. 5, Supplementary Tables 2 and 3).

**Oxytocin/Vasorpessin receptor family subtype specificity.** OT differs only in two positions from the closely related AVP (Ile[3] and Leu[8] in OT *vs.* Phe[3] and Arg[8] in AVP) (Fig. 4a). While these differences suffice to render OT specific for the OTR over the vasopressin receptors, the OTR itself is not selective between OT and AVP[22]. To investigate possible contributions to subtype selectivity, we compared the OT-bound OTR structure to the previously published AVP-bound $V_2R$ structures[21,23,24] (Fig. 4). Both OT and AVP adopt a similar orientation in the orthosteric pocket of their respective receptor with highest similarity observed for the agonist's cyclic portion, where Phe[3] of AVP penetrates only marginally deeper than Ile[3] of OT (Fig. 4). The largest structural differences are found for positions eight and

nine of the ligand in the tripeptide C terminus. Leu[8] and Gly[9] in OT are located along the ring plane, with Gly[9] facing helix I (Figs. 1d and 4). Conversely, Arg[8] and Gly[9] in AVP, which in each reported $V_2R$:AVP structure have been modelled differently, are facing away from the ring plane, enabling contacts to residues of ECL1, ECL3 and the N terminus of $V_2R$. In the OTR, helix I adopts a position further away from the central axis of the receptor compared to $V_2R$. This wider opening of the orthosteric pocket in OTR enables binding of the OT C-terminal residues Leu[8] and Gly[9] in the observed conformation. In $V_2R$, however, OT binding would be sterically compromised due to a clash of Gly[9] with helix I. In contrast, the bound conformation of AVP in $V_2R$ is compatible with binding to OTR. Therefore, the positioning of helix I might explain why the OTR is not selective between OT and AVP, but OT binding is specific to OTR and it does not bind to $V_2R$. In $V_2R$, helix I is packed in a more compact manner in the helix bundle compared to the OTR, so it would clash with Gly[9] if OT were to be expected to adopt the identical binding conformation as observed in complex with OTR.

In variance to long-standing models[22,25], we do not observe an interaction of OT with R34[N terminus] or F103[ECL1] of the OTR. This might indicate an allosteric effect, or in the case of R34, which does not show clear density, a more dynamic interaction. We speculate that this could additionally contribute to subtype selectivity in the OT/AVP receptor family.

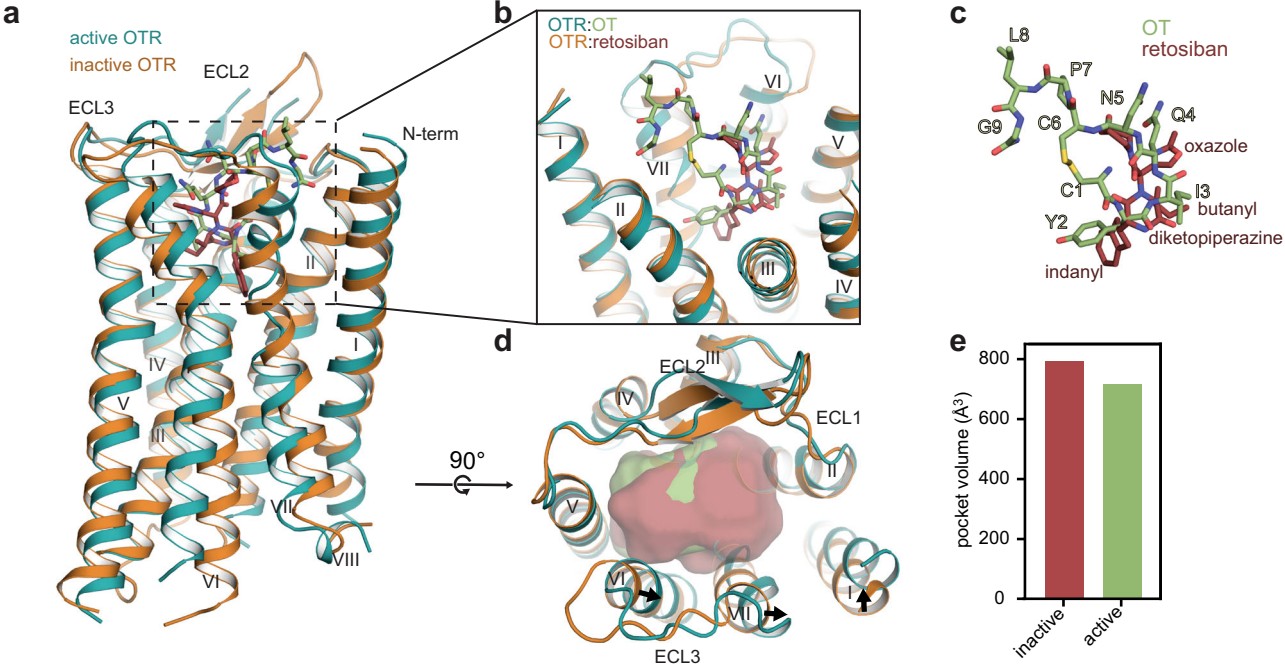

**Fig. 2 Ligand-induced changes in OTR activation. a** Superposition of active OTR:OT complex (teal) and inactive OTR:retosiban complex (orange, PDB ID: 6TPK). **b** Close-up on binding pockets of OT and retosiban viewed from the extracellular side. **c** OT and retosiban binding modes as viewed from the membrane plane. **d** Extracellular view of the super-positioned receptors with calculated pocket volume shown as surface representation. Arrows indicate shifts of the extracellular helix tips from inactive to active state. **e** Calculated pocket volumes for inactive and active OTR conformations. Pocket volumes were calculated with POVME 2.0[42].

On the intracellular receptor side, the main conformational differences between OTR and $V_2R$ are found in the positioning of the helix VII-VIII transition region and the elongation of helix V of $V_2R$ compared to OTR (Supplementary Fig. 6). Both regions contribute to G protein binding, and the respective differences are likely a feature of the diverse binding modes between the receptors and the interacting Gα subunits ($V_2R:G_s$ and OTR:$G_{o/q}$).

**G protein interaction.** In the signalling complex of the activated OTR, the α5 helix of $G_{o/q}$ is bound in a crevice constituted by helices II, III, V, VI and VII at the receptor intracellular side, where the G protein C-terminal residues E350 and Y351 engage in hydrogen bonding with R73$^{2.38}$ and D136$^{3.49}$ of OTR (Fig. 5a, d). Compared to reported receptor:$G_{s/q}$ complexes[12,26,27], the $G_q$ α5 helix in OTR:$G_{o/q}$ is rotated away from helix VI (Fig. 5b). This rotation cannot be attributed to differences in chimera design, as there is a perfect structural alignment of the individual $G_α$ subunits (Fig. 5e). Most importantly, the same rotation is also distinct from GPCR:$G_o$ structures[28–30] (Fig. 5c). Instead, the orientation of the $G_q$ α5 helix in OTR:$G_{o/q}$ very much resembles that of $G_s$ coupled to $V_2R$, with the latter penetrating less deeply into the $V_2R$ intracellular TMD crevice, suggesting the receptor is governing the α5 orientation (Fig. 5f).

In conclusion, we report here the structure of the human OTR:OT signalling complex, providing insights into the subfamily-specific OTR activation mechanism and an unexpected G protein binding mode as well as the detailed OT binding mode, thereby enhancing our understanding of the subtype specificity within the closely related oxytocin and vasopressin receptor family. After the present manuscript had been submitted, a related structure with a different G protein has appeared, reaching similar conclusions[31]. Together, these findings are expected to greatly facilitate the development of novel therapeutics for the treatment of a variety of OTR-implicated diseases.

## Methods

**Design of complex constructs**. The sequences of scFv16[14] and of wild-type human OTR (wtOTR), codon-optimised for expression in *Spodoptera frugiperda* (*Sf*9) (C-terminally truncated after residue 359), were cloned into a modified pFL vector (MultiBac system, Geneva Biotech) for *Sf*9 expression. The resulting expression constructs contained a melittin signal sequence, followed by a FLAG-tag, a His$_{10}$-tag, and a human rhinovirus 3C protease cleavage site N-terminal to the gene of interest. To increase OTR purification yield, the mutation D153Y was introduced into the truncated wtOTR sequence, as identified previously[10]. It shows very similar KD and EC50 for oxytocin (OT) (Supplementary Fig. 1). The mutant was generated by sequence- and ligation-independent cloning as previously described in detail[32]. To generate an Gα$_{o/q}$ subunit that would allow interaction with scFv16[30], the N-terminal 18 amino acids of G$_{i1}$ were introduced to the engineered mini-G$_o$12[11]. To generate G$_q$-like interactions, residues H5.16, H5.17, H5.18, H5.22, H5.23, H5.24, and H5.26 (according to the common Gα numbering)[33] in the C-terminal helix were mutated to corresponding amino acids of G$_q$. Finally, the engineered Gα$_{o/q}$ chimera sequence was cloned into one pFL vector, together with Gβ$_1$ (including an N-terminal non-cleavable His$_{10}$-tag) and with Gγ$_2$, with each gene under the control of its own polyhedrin promoter.

**Expression and purification of OTR**. Expression and purification were performed as previously described in detail[12,17]. In brief, 4 L of *Sf*9 insect cell culture at a density of $3 \times 10^6$ cells/ml were infected with baculovirus stocks at a multiplicity of infection ≥5. Cells were harvested 72 h post-infection and stored at −80 °C. Cells were thawed on ice before purification, lysed, and membranes were isolated by repeated Dounce homogenisation in hypotonic buffer containing 10 mM Hepes (pH 7.5), 20 mM KCl, 10 mM MgCl$_2$, and protease inhibitors (50 µg/ml Pefabloc SC and 1 µg/ml Pepstatin A, both Carl Roth) and then in hypertonic buffer containing 10 mM Hepes (pH 7.5), 20 mM KCl, 10 mM MgCl$_2$, 1.0 M NaCl, 0.1 mg/ml deoxyribonuclease (DNase, MilliporeSigma) and protease inhibitors. Washed membranes were resuspended in hypotonic buffer, and the low-affinity antagonist SSR 149415 (Tocris) was added to a final concentration of 100 µM and the suspension was incubated for 30 min. Then, iodoacetamide (2 mg/ml final concentration; MilliporeSigma) was added to the solution followed by another 30 min of incubation. Subsequently, the receptor was solubilized in buffer containing 30 mM Hepes (pH 7.5), 500 mM NaCl, 10 mM KCl, 5 mM MgCl$_2$, 50 µM SSR 149415, 1% (w/v) n-dodecyl-β-d-maltopyranoside (DDM, Anatrace), and 0.2% (w/v) cholesteryl hemisuccinate (CHS, MilliporeSigma) for 3 h at 4 °C. Insoluble material was removed by ultracentrifugation at 220,000 *g*, and the supernatant containing the solubilized receptor was incubated with TALON IMAC resin (Cytiva) at 4 °C.

The receptor-bound resin was washed with 20 column volumes (CVs) of wash buffer I containing 50 mM Hepes (pH 7.5), 500 mM NaCl, 10 mM MgCl$_2$, 5 mM

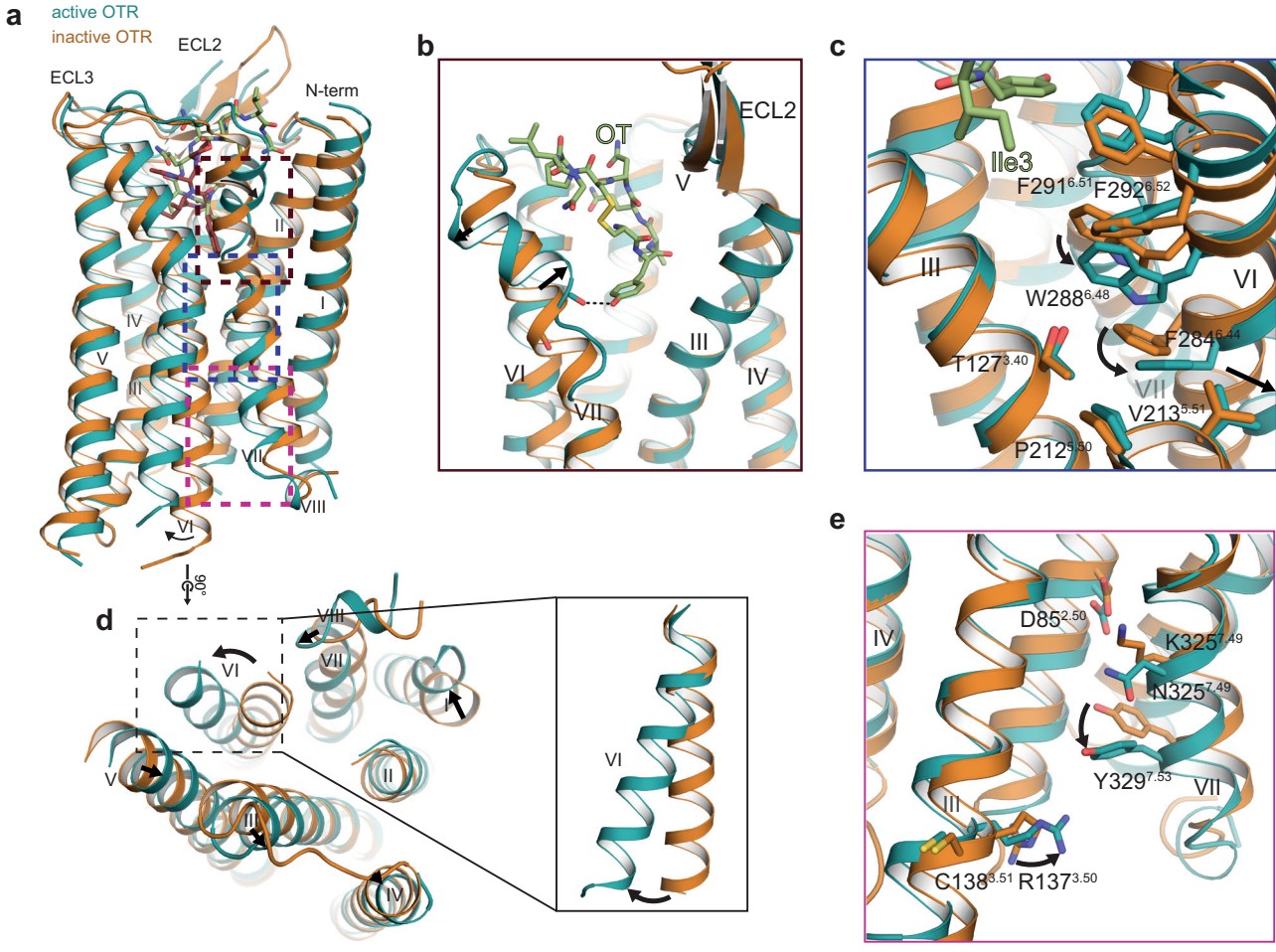

**Fig. 3 Activation mechanism of OTR. a** Molecular mechanism of OTR activation. Superposition of active (teal, this study; PDB 7QVM) and inactive OTR (orange, PDB 6TPK). **b** Close-up on kink in helix VII. Hydrogen bond between OTR and OT is indicated by dashed black line. **c** Close-up view on class A-specific microswitch motifs CWxP and PIF. Arrows indicate shifts of microswitch residues from inactive to active state. **d** Intracellular view of super-positioned receptors with additional close-up view on helix VI. Arrows indicate shifts of the intracellular helix tips from inactive to active state. **e** Close-up view on class A-specific microswitch motifs DRC and NPxxY. Arrows indicate shifts of microswitch residues from inactive to active state.

imidazole, 10% (v/v) glycerol, 8 mM adenosine triphosphate (ATP, MilliporeSigma), supplemented with 0.5% (w/v) DDM, 0.01% (w/v) CHS, and 10 µM SSR 149415. Subsequently, the detergent was exchanged with 16 CVs of wash buffer I supplemented with 1% (w/v) lauryl maltose neopentyl glycol (LMNG, Anatrace), 0.1% (w/v) CHS, and 10 µM SSR 149415, and the antagonist was replaced with OT (Psyclo Peptide Inc.) with another 16 CVs of wash buffer I supplemented with 0.5% (w/v) LMNG, 0.05% (w/v) CHS, and 100 µM OT. Agonist-bound OTR was finally washed with 25 CVs of wash buffer II containing 50 mM Hepes (pH 7.5), 150 mM KCl, 15 mM imidazole, 10% (v/v) glycerol, 0.05% (w/v) LMNG, 0.005% (w/v) CHS, and 50 µM OT, before eluting the receptor in 4 subsequent fractions of 1 CV of elution buffer containing 50 mM Hepes (pH 7.5), 150 mM KCl, 250 mM imidazole, 10% (v/v) glycerol, 0.01% (w/v) LMNG, 0.001% (w/v) CHS, and 50 µM OT.

Protein-containing fractions were combined and concentrated to 0.5 ml using a 50-kDa molecular weight cutoff (MWCO) Vivaspin 2 concentrator (Sartorius), and desalted by buffer exchange on a PD MiniTrap G-25 column (Cytiva) equilibrated with G25 buffer containing 50 mM Hepes (pH 7.5), 150 mM KCl, 10% (v/v) glycerol, 0.01% (w/v) LMNG, 0.001% (w/v) CHS, and 50 µM OT. To remove the N-terminal affinity tags and to deglycosylate the receptor, OT-bound receptor was treated with His-tagged 3 C protease and His-tagged peptide N-glycosidase (PNGase) F (both prepared in-house) overnight. To collect cleaved receptor, the reaction was incubated with Ni-nitrilotriacetic acid (Ni-NTA) resin (Cytiva) for 1 h, cleaved receptor was collected as the flow-through, then concentrated to ~3 to 5 mg/ml with a 50-kDa MWCO Vivaspin 2 concentrator, and directly used for complex formation. Protein purity and monodispersity were assessed by LDS–polyacrylamide gel electrophoresis and analytical size exclusion chromatography (SEC) using a Nanofilm SEC-250 column (Sepax).

**Purification of G$_{o/q}$.** Purification of the engineered heterotrimeric G protein was carried out similarly to receptor purification, with small adaptations. All buffers used were supplemented with 10 µM guanosine diphosphate (GTP,

MilliporeSigma) and 100 µM tris(2-carboxyethyl)phosphine) (TCEP, Thermo Fisher Scientific). In contrast to the receptor purification, monovalent cation concentration never exceeded 150 mM, and all buffers were devoid of any receptor ligands and iodoacetamide. Enriched G protein-containing membranes were washed by Dounce homogenisation without high salt concentrations in physiological buffer, containing 10 mM Hepes (pH 7.5), 150 mM NaCl, 20 mM KCl, 10 mM MgCl$_2$, and protease inhibitors. Solubilisation and immobilisation on TALON IMAC resin was performed as described above.

The G protein–bound resin was initially washed with 30 CVs of wash buffer I containing 50 mM Hepes (pH 7.5), 150 mM KCl, 10 mM MgCl$_2$, 5 mM imidazole, 10% (v/v) glycerol, 1% (w/v) DDM, 0.2% (w/v) CHS, 10 µM GTP and 100 µM TCEP followed by a wash with 30 CV of detergent exchange buffer containing 50 mM Hepes (pH 7.5), 150 mM KCl, 1 mM MgCl$_2$, 5 mM imidazole, 10% (v/v) glycerol, 1% (w/v) LMNG, 0. 1% (w/v) CHS 8 mM ATP, 10 µM GTP and 100 µM TCEP. G proteins were finally washed with 30 CVs of wash buffer II containing 50 mM Hepes (pH 7.5), 150 mM KCl, 1 mM MgCl$_2$, 15 mM imidazole, 10% (v/v) glycerol, 0.01% (w/v) LMNG, 0.001% (w/v) CHS, 10 µM GTP and 100 µM TCEP and eluted stepwise in 4 subsequent fractions of 1 CV of elution buffer containing 750 mM imidazole.

Eluted G protein was concentrated using a 100-kDa MWCO Vivaspin 2 concentrator and further purified using a Superdex 200 10/300 (Cytiva) column equilibrated with SEC buffer containing 25 mM Hepes (pH 7.5), 150 mM KCl, 2% (v/v) glycerol, 0.01% (w/v) LMNG, and 0.001% (w/v) CHS. Fractions containing monodisperse G protein were collected, concentrated to 7.1 mg/ml with a 100-kDa MWCO Vivaspin 2 concentrator (Sartorius), frozen in liquid nitrogen, and stored at −80 °C until use.

**Purification of scFv16.** ScFv16 was expressed and purified as described before[12]. Briefly, scFv16 was expressed by secretion from baculovirus-infected Sf9 cells for 72 h. ScFv16-containing expression medium was separated from cells by

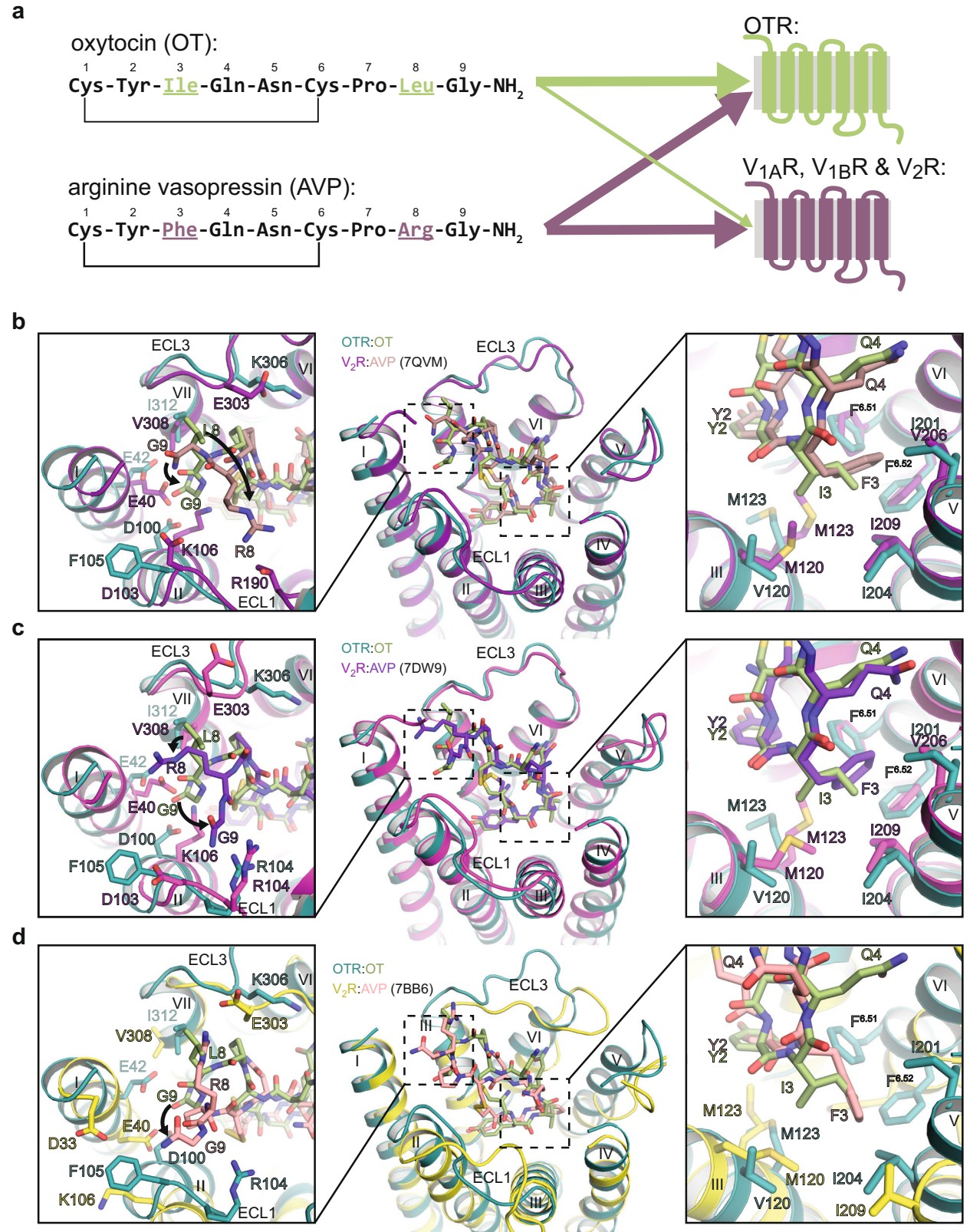

**Fig. 4 Comparison of the OTR and V₂R orthosteric binding pockets. a** (left) Amino acid sequences of OT and AVP. Amino acid differences between the closely related hormones are highlighted. (right) Simplified specificity profile of OT and AVP for oxytocin and vasopressin receptors. Line thickness indicates affinity towards indicated receptors. **b–d** Structural superposition of OTR:OT with V₂R:AVP structures (**b** PDB ID: 7QVM; **c** PDB ID: 7DW9; **d** PDB ID: 7BB6), illustrating the significant differences between AVP positions 3 and 8 in the two reported V₂R:AVP structures. Arrows indicate conformational changes in non-conserved positions of AVP and OTR. (left) Close-up on sub-pocket binding AVP/OTR position 8. (middle) Overview of binding pockets. (right) Close-up on sub-pocket binding AVP/OTR position 3.

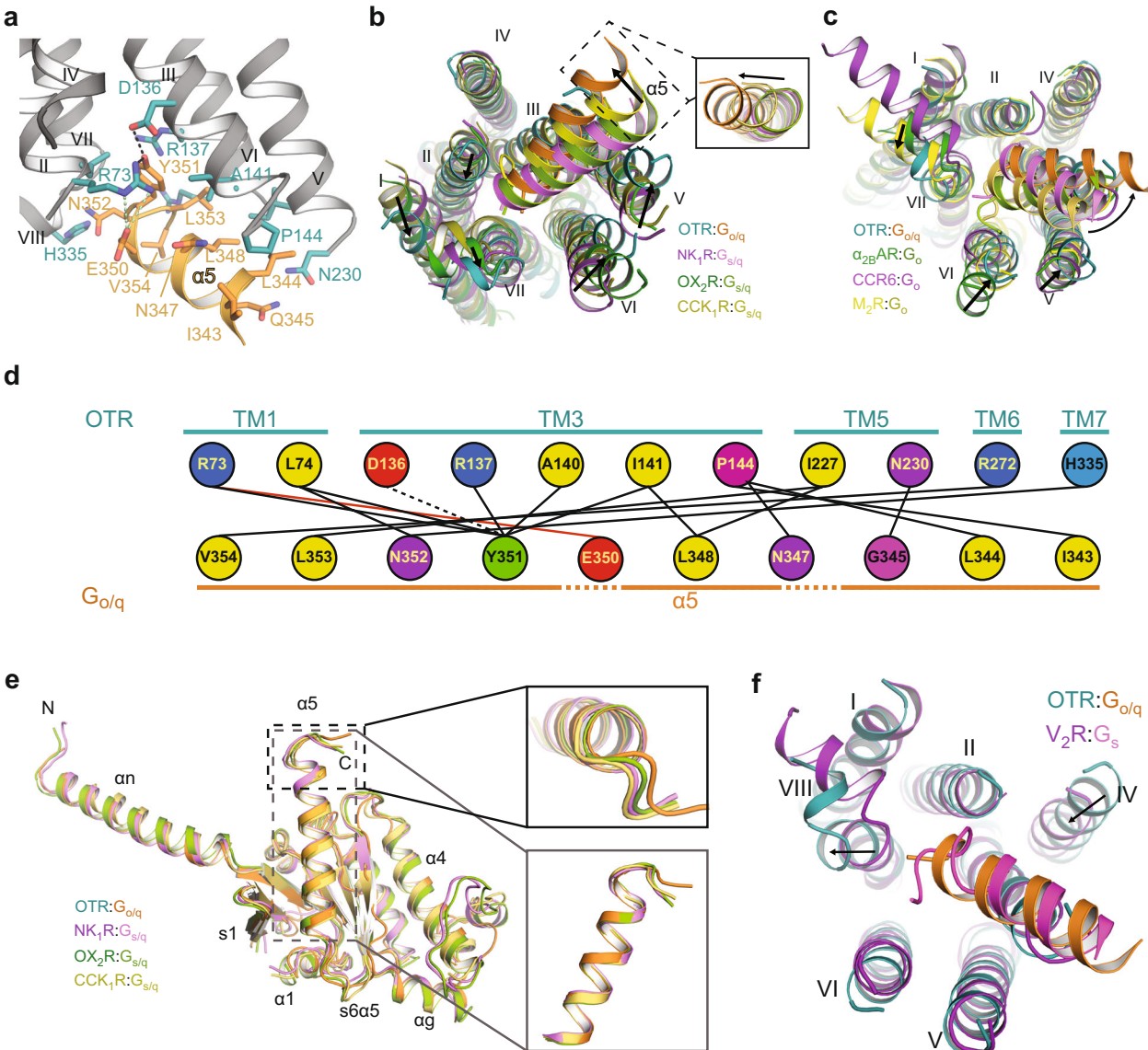

**Fig. 5 Interactions of OTR and Gα$_{o/q}$. a** Detailed interactions of G$_q$ α5 with OTR. **b** Structural comparison of OTR:G$_{o/q}$ to G$_{s/q}$ complexes with focus on G$_q$ α5 positioning: NK$_1$R:G$_{s/q}$ (purple, PDB 7P00), OX$_2$R:G$_{s/q}$ (green, PDB 7L1U), and CCK$_1$R:G$_{s/q}$ (yellow, PDB 7MBY). **c** Structural superposition of OTR with α$_1$AR:G$_o$ (green; PDB ID: 6K41), CCR6:G$_o$ (purple; PDB ID: 6WWZ), and M$_2$R:G$_o$ (yellow; PDB ID: 6OIK) as viewed from the intracellular side with a focus on receptor – G protein interaction. Arrows indicate structural differences between intracellular receptor helix tips and α5 orientations. **d** Schematic drawing of direct interactions between OTR and the α5 helix of Gα$_{o/q}$. Hydrogen bonds are drawn as dashed lines. Salt bridges are indicated by lines coloured in red. Amino acids are coloured according to their biophysical properties. **e** Structural superposition of OTR-bound G$_{o/q}$ with G$_{s/q}$ from different signalling complexes: NK$_1$R:G$_{s/q}$ (purple; PDB ID: 7P00), OX$_2$R:G$_{s/q}$ (green; PDB ID: 7L1U), and CCK$_1$R:G$_{s/q}$ (yellow, PDB ID: 7MBY) with two different close-up views on Gα5. **f** Structural comparison of OTR:G$_{o/q}$ and V$_2$R:G$_s$ (pink, PDB 7KH0) with focus on helix α5 positioning.

centrifugation, then pH-adjusted by addition of Hepes (pH 7.5). Metal-chelating agents from the cells and medium were quenched by incubation with 1 mM CoCl$_2$ and 5 mM CaCl$_2$ for 1 h at 22 °C. Resulting precipitates were removed by centrifugation, and the filtrated supernatant was loaded onto a Co$^{2+}$-loaded HiTrap IMAC HP column (Cytiva). The column was washed with 20 CVs of buffer A containing 20 mM Hepes (pH 7.5), 150 mM NaCl, 1 mM MgCl$_2$, 4 mM ATP, and 5 mM imidazole, followed by 20 CVs of buffer B containing 20 mM Hepes (pH 7.5), 150 mM NaCl, and 30 mM imidazole. The protein was eluted with buffer C [20 mM Hepes (pH 7.5), 150 mM NaCl, and 300 mM imidazole]. Monomeric fractions were pooled, concentrated using a 10-kDa MWCO Amicon Ultra concentrator (Merck), and imidazole was removed by applying the concentrate to a PD-10 desalting column (Cytiva) equilibrated with G25-buffer containing 50 mM Hepes (pH 7.5), 150 mM NaCl, 10% (v/v) glycerol, 0.01% (w/v) LMNG, and 0.001% (w/v) CHS. ScFv16 was incubated for 3 h with His-tagged 3 C protease and His-tagged PNGase F for removal of affinity tags and deglycosylation. After incubation with Ni-NTA resin for 1 h, cleaved scFv16 was collected as the flow-through, concentrated with a 10-kDa MWCO Amicon Ultra concentrator, and further purified using a Superdex 200 10/300 column equilibrated with SEC buffer.

Monomeric fractions were pooled, concentrated to ~6 to 7 mg/ml, flash-frozen in liquid nitrogen, and stored at −80 °C. Right before complex formation, scFv16 was thawed and the buffer was exchanged on a PD MiniTrap$^{TM}$ G-25 column (Cytiva) equilibrated with G25-buffer.

**Complex formation.** Purified OTR and engineered heterotrimeric G protein (Gα$_{o/q}$Gβ$_1$Gγ$_2$) were mixed in a molar ratio of 1:1.2 in complexation buffer (25 mM Hepes (pH 7.5), 100 mM KCl, 1 mM MgCl$_2$, 0.01% (w/v) LMNG, 0.001% (w/v) CHS, 100 µM OT, and 100 µm TCEP). After 30 min, apyrase (0.8 U/ml; Milli-poreSigma) and λ-phosphatase (1,000 U/ml, New England Biolabs) were added to the mixture. After 2 h, purified scFv16 was added at 4-fold molar excess over receptor, and complex formation was allowed to proceed overnight at 4 °C. Stable complex was isolated by SEC on a Superdex 200 10/300 column (Cytiva) equilibrated with blotting buffer (25 mM Hepes (pH 7.5), 100 mM KCl, 1 mM MgCl$_2$, 100 µM OT, and 100 µm TCEP, 0.001% (w/v) LMNG, and 0.0001% (w/v) CHS). Corresponding peak fractions were concentrated to 1 mg/ml for EM studies, using a 100-kDa MWCO Vivaspin Turbo PES (Sartorius) concentrator.

**Single-particle cryo-EM data collection**. For cryo-EM grid preparation, 3 μL of purified OTR:OT:G$_{o/q}$:scFv16 complex in blotting buffer were applied to glow-discharged holey carbon gold grids (Quantifoil R1.2/1.3, 300 mesh), and subsequently vitrified using a Vitrobot Mark IV (Thermo Fisher Scientific) operated at 100% humidity and 4 °C. Cryo-EM images were acquired by a Titan Krios G3i electron microscope (Thermo Fisher Scientific), operated at 300 kV, at a nominal magnification of 130,000 using a K3 direct electron detector (Gatan) in super-resolution mode, corresponding to a pixel size of 0.325 Å. A BioQuantum energy filter (Gatan) was operated in a zero-loss mode, using 20 eV energy slit-width. A total of 6,450 movies were obtained, with a defocus range of −0.8 to −2.4 μm using automatic data acquisition with EPU software (version 2.5;Thermo Fisher Scientific). The total exposure time was 1.79 s with an accumulated dose of ~63.78 electrons/Å$^2$ and a total of 67 frames per micrograph. A second set of 5,217 image stacks were acquired with the same conditions and parameters.

**Single-particle cryo-EM data processing**. All image stacks were binned to generate a pixel size of 0.65 Å followed by motion-correction and dose-weighting using MotionCor2[34] (version 1.4). All images were contrast transfer function corrected using Gctf[35] (version 1.06), as implemented in cryoSPARC[36](version 3.0.1). Subsequent image processing steps were performed in cryoSPARC. Initial particle selection was done using the automated blob picker on 100 micrographs, using a 100 Å minimum and 150 Å maximum particle diameter, to extract a total of 38,166 particles. Next, the extracted particles were subjected to one round of 2D classification, into 200 classes, from which 7 classes were selected and used as a template for the automatic particle picking process. A total of 3,062,337 particles were extracted from the first 6,450 micrographs, followed by a round of 2D classification that resulted in 200 classes. Finally, a round of 3D reconstructions and classification produced 6 classes.

A total of 3,504,800 particles were extracted from the second data-set of image-stacks, followed by a round of 2D classification, split into 200 classes, and a round of 3D reconstructions and classification into 3 classes. The particles from the best classes from both datasets were then joined together and subjected to one ab-initio round of 3D reconstructions split into 6 classes. A final data-set of 392,370 particles from the best 3D classes were subjected to local and global CTF refinements, followed by a 3D non-uniform refinement. The final density map was resolved to 3.25 Å, after map sharpening, as determined by gold-standard Fourier shell correlation using the 0.143 criterion. Local resolution estimation was performed using cryoSPARC.

**Model building**. An initial model was created by docking the individual complex components into the cryo-EM map using the "fit in map" routine in UCSF Chimera[37] (version 1.15). The following structures from the Protein Data Bank (PDB) were used: OTR (PDB ID: 6TPK), Gα$_o$ (PDB ID: 6WWZ), Gβ$_1$γ$_2$, scFv16 (PDB ID: 6OIJ), and AVP (PDB ID: 7DW9). All initial model components were manually rebuilt in Coot[38] (version 0.9.7), followed by several rounds of manual real-space refinement in Coot and real-space refinement with the software package Phenix.real_space_refine in Phenix[39] (version 1.20.1-4487). The quality of the models was assessed using MolProbity[40] before PDB deposition. PyMOL (version 2.5) was used for visual inspection, model comparison and figure preparation.

**IP1 accumulation assays**. Ligand-induced IP1 accumulation and ligand-binding experiments were measured using transiently transfected Human Embryonic Kidney (HEK) 293 T/17 cells (American Type Culture Collection). The cells were cultivated in Dulbecco's modified Eagle's medium (Thermo Fisher Scientific) supplemented with penicillin (100 U/ml), streptomycin (100 μg/ml, Milli-poreSigma), and 10% (v/v) foetal calf serum (BioConcept) and maintained at 37 °C in a humidified atmosphere of 5% CO$_2$ and 95% air. Transient transfections were performed with TransIT-293 (Mirus Bio) according to the manufacturer's instructions. Full-length OTR, truncated V$_{1A}$R (residues 1-378 with C-terminal mutation T378K) and mutants thereof were directly cloned into a mammalian expression vector containing an N-terminal SNAP-tag (pMC08; Cisbio). Cells were transfected and directly seeded at 7500 cells per well in poly-D-lysine-coated white 384-well plates (Greiner).

To compare IP1 accumulation a homogeneous time-resolved fluorescence (HTRF) signalling assay was performed, adapting a previously described protocol[41]. The cells were washed 48 h after transfection with phosphate-buffered saline (PBS) and stimulation buffer (Cisbio) and subsequently incubated for 1 h at 37 °C with a concentration range of oxytocin (Psyclo Peptide Inc.) diluted in stimulation buffer. The IP1 accumulation was determined using the HTRF IP-One Kit (Cisbio) according to the manufacturer's protocol. Fluorescence intensities were measured on a Spark fluorescence plate reader (Tecan). To generate concentration-response curves, data were analysed by a three-parameter logistic equation in GraphPad Prism software (version 9.2.0).

**Whole-cell ligand binding assays**. Ligand-binding experiments were performed on whole cells for comparison of affinities for wild-type and receptor mutants using an HTRF binding assay as previously described[17]. Forty-eight hours post-transfection the cells were labelled with 50 nM SNAP-Lumi4-Tb (Cisbio) in labelling buffer (20 mM Hepes (pH 7.5), 100 mM NaCl, 3 mM MgCl$_2$, and 0.05% (w/v) bovine serum albumin (BSA)) for 1.5 h at 37 °C. The cells were washed two times with labelling buffer and two times with assay buffer (20 mM Hepes (pH 7.5), 100 mM KCl, 3 mM MgCl$_2$, and 0.05% (w/v) BSA)) and subsequently incubated for 1 h at room temperature with a concentration range of fluorescently labelled peptide HiLyte Fluor 488-Orn[8] oxytocin (Eurogentec) in assay buffer. Fluorescence intensities were measured on a Spark fluorescence plate reader with an excitation wavelength of 340 nm and emission wavelengths of 620 nm and 510 nm for Tb$^{3+}$ and the fluorophore HiLyte Fluor 488, respectively. The ratio of fluorescence resonance energy transfer (FRET) donor and acceptor fluorescence intensities was calculated (F510 nm/F620 nm). Nonspecific binding was determined in the presence of a 1000-fold excess of unlabelled oxytocin. Data were analysed by global fitting to a one-site saturation binding equation with the GraphPad Prism software.

**Reporting summary**. Further information on research design is available in the Nature Research Reporting Summary linked to this article.

## Data availability

Atomic coordinates of the OTR:G$_{o/q}$:OT:scFv16 complex have been deposited in the PDB under the accession code 7QVM. Cryo-EM maps used have been deposited in the EMDB found under code EMD-14180. Source data for ligand binding and receptor activation are provided with this paper.

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

## Acknowledgements
We thank the staff, in particular S. Sorrentino, of the Centre for Microscopy and Image Analysis at the University of Zurich for their support in the initial phases of the project. We further thank O. Eggenberger for assistance in protein production and I. Berger for providing us with baculovirus transfer vectors. This work was supported by Schweizer-ischer Nationalfonds Grant 31003A_182334 (to A.P), and by the European Research Council (810057-HighResCells to A.P. and O.M.).

## Author contributions
The project was outlined by Y.W. and A.P. Y.W. and J.E. designed the G protein and scFv constructs. Y.W. purified all proteins and prepared the complexes. Y.W, J.E. and S.V. vitrified the samples on cryo-EM grids. Y.W. and J.E. collected cryo-EM data with the help of S.V. Y.W. and S.V. processed data and refined the cryo-EM density map. J.E. and Y.W. built and refined the structure model. C.T. designed and performed the functional assays. Project management was carried out by Y.W., J.E., C.T., O.M. and A.P. The manuscript was prepared by Y.W., C.T., S.V., J.E., O.M. and A.P. All authors contributed to the final editing and approval of the manuscript.

## Competing interests
The authors declare no competing interests.
