## [Peer Review File · Nature Communications]

Structural basis for the activation and ligand recognition of the human oxytocin receptorREVIEWERS' COMMENTS

Reviewer #1 (Remarks to the Author):

The manuscript by Waltenspühl Y et al. describes the three-dimensional structure of the active human oxytocin (OT) receptor (OTR) in complex with its natural ligand oxytocin, a chimeric mini-Go/q/i protein, and the stabilizing antibody fragment ScFv16. The study enabled single-particle cryo-electron microscopy (cryo-EM) analysis of the OT:OTR:Go/q/i:ScFv16 complex at a resolution of 3.2 Å allowing to provide a structural basis for the activation and ligand recognition of this G protein-coupled receptor (GPCR) which plays a central role in the regulation of social behavior and reproduction.

This work is particularly important because it is complementary to the publication of the crystal structure of an antagonist-bound OTR by the same research team 2 years ago (the antagonist being the nonpeptide retosiban). Interestingly, the comparison between the inactive and active states of the receptor is quite useful to increase our knowledge of molecular determinants involved in its activation mechanism. In the future, this structural information will also help in the rational design and development of novel OT-based ligands for treating socio-behavioral pathologies.

The expertise of the authors in the scientific field of GPCRs and structural biology is internationally well renowned, the methods developed are state-of-the-art and the results presented in the article are convincing. In addition, the study is in total agreement with those describing the active structures of the OTR-related arginine-vasopressin (AVP) V2 receptor (V2R) in complex with AVP and its canonical Gs signaling protein, that were published last year. However, I have some criticisms about the organization of the manuscript and I also have some minor comments.

1/ major comment:

I would re-organize the article for several reasons. First of all, I do not understand why only 2 principal figures are included in the main document. Second, some panels, for instance those in figure 2b and supplementary figure 7, are shown twice. Third, some close-up illustrations do not correspond to their corresponding squared dashed lines. Finally, some data illustrated in the figures are poorly discussed. I think the format of Nature Communications Journal is quite flexible and because the present document is short, the main text may be lengthened.

Here are my proposals and this will change of course the numbering and organization of the different principal figures:

-I agree with principal figure 1 which illustrates the structure of the complex, the ligand-binding site, and OT binding pose.

-Then, I would compare active to inactive structures of OTR to describe the superposition of OT-retosiban and volume receptor binding pocket volumes. The new principal figure 2 would be the current extended data figure 5, panels a-e.

-Then, I would still compare active to inactive structures of OTR to discuss molecular rearrangements and conservation of activation mechanisms in the AVP/OT receptor family. The new principal figure 3 would be current principal figure 2a and current extended data figure 5 f-g. Current extended data figure 6 will become extended data figure 5.

-Then I would present the G protein binding interface and compare it to other GPCR-G protein interfaces. The new principal figure 4 would be a mix of current extended data figure 8 and current principal figure 2, panels c-e.

-Finally, I would detail receptor subtype specificity in the AVP/OT family. This would be illustrated by the current extended data figure 7 that will become the new principal figure 5. In addition, because panel 2b is equivalent to different panels shown in the current extended data figure 7, it may be canceled.

Five principal figures instead of two may be more appropriate for this article format. In agreement, only 5 extended data figures would be necessary.

2/ minor comments:

Some references are missing in the bibliography.

a) In the main text (page 3, second paragraph, line 32), while the authors claim that OT-based treatments of socio-behavioral deficiencies have yet failed, they may add a reference about the first nonpeptide OTR agonist that improves social interaction in a mouse model of autism. Indeed, LIT-001 can penetrate the blood-brain barrier and displays potential advantages of structural simplicity, low molecular weight, and drug-likeness as compared to OT (Frantz MC et al., *J Med Chem*, 2018, 61(19), 8670-92). However, indeed, this compound is not highly selective to OTR.

b) In the main text, pages 4 and 5, while describing the OT binding pose in the receptor orthosteric binding site, the authors should acknowledge that their results are in accordance with previous data. AVP and OT are two chemically-related hormones (two cyclic nonapeptides with only 2 different residues at positions 3 and 8) and their targets constitute a high-homology receptor family sharing a common conserved binding site. At least, the authors should cite the first experimentally-validated model of AVP binding in V1aR published in 1995 (Mouillac B et al., *J Biol Chem*, 270(43), 25771-77), since the identification of receptor residues involved in the binding of both hormones in the two studies is consistent. Hopefully, more recent papers (Bous J et al., *Sci Adv*, 2021, 7, eabg5628; Zhou F et al., *Cell Res*, 2021, 31, 929-31; Wang L et al., *Cell Res*, 2021, 31, 932-34) where cryo-EM active structures of AVP-bound V2R in complex with Gs protein have been determined, are currently included in the reference list.

c) A paper by Meyerowitz JG et al., very recently accepted for publication in *Nature Structural Molecular Biology* (it can be seen in the latest research articles, 2022, March 3rd, on the web page of this journal), should be added as a reference. In this article, the cryo-EM structure of the OT-bound OTR receptor in complex with a mini-Gq/i (at a resolution of 2.9 Å) is described. It is in complete agreement with that described in the present paper by Waltenspühl et al. Interestingly, the two papers are complementary. This study may be cited on page 4, lines 58-59. The authors should indicate that their study, in

agreement with that by Meyerowitz and colleagues, enabled unprecedented insights into OTR activation by OT and into OTR-G protein interaction.

Reviewer #2 (Remarks to the Author):

The authors present a crystal structure of the Oxytocin receptor (OXTR), at a resolution of 3.2 Å, which, is not particularly high in the context of the most recent solved membrane protein structures. Recently, the same authors reported another OXTR structure (Crystal structure of the human oxytocin receptor. Waltenspühl Y, Schöppe J, Ehrenmann J, Kummer L, Plückthun A. *Sci Adv.* 2020). The previous structure was solved by employing a small non peptidic antagonist (retosiban), which enabled the stabilization of a presumably inactive receptor conformation. Not surprisingly, the orthosteric binding site defined in the present paper overlaps with the one already described in the previous structure and with previous models obtained by homology modelling and site directed mutagenesis. Shifts of extracellular and intracellular helix tips and Class-A specific microswitches (in the CWxP, PIF, NPxxY, DRY) from inactive to active states are reported in extended data fig.5, with no major breakthrough. Several residues described here as relevant for binding have been already subjected to site-directed mutagenesis, binding and activation studies, and it would have been appropriate to cite and discuss this literature (which the authors could easily find and access). As an example, concerning the discussion of “Early functional studies on OT derivatives” (line 98), the authors should refer to the extensive work reported by Berde and Boissonnas (In Berde, B. (ed.), 1968, *Handbook of Experimental Pharmacology. Neurohypophysial Hormones and Similar Polypeptides.* Springer-Verlag) in which substitutions of all residues of OT and AVP with a number of L- and D- aminoacids (including O-MeTyr at position 2) were reported; this work remains a milestone in the field and constitutes the ground for the following years of OT/AVP peptide design and synthesis (see for example Hruby and Chow, M.S. (1990) *Annu. Rev. Pharmacol. Toxicol.*; Hruby and Smith (1987) In Smith, C.W. (ed.), *The Peptides. Chemistry, Biology and Medicine of Neurohypophysial Hormones and Their Analogs.* Academic Press). In particular, please note that alkylation of residue 2 was already reported in 1968 by Berde, B. and Boissonnas, R.A to confer antagonistic properties to OT/AVP ligands and extensively exploited to design potent and selective OXTR antagonists (Manning and Sawyer, *J Rec Res* 1993). I suggest that integrating their actual data with the data already present in the extensive OT/AVP literature could help to identify new clues for the development of OT-based therapeutics, as stated in the conclusion, improving the interest of the manuscript.

This also apply to the discussion of the OTR/V2 structure, as many ligands display selective agonist/antagonists (biased agonists profiles at these receptors: could the authors provide new clues for the design of more selective/potents analogs?

Just two minor points

Lines 132-133 The sentence “in contrast, the binding conformation of AVp in V2R is compatible with OTR binding” is not clear to me. Please explain

Lines 135-139: A direct interaction between F103 in the OTR and OT has never been proposed, to my knowledge: an interaction was however proposed at D and Y residues present at that position in V2 and V1a receptors to explain the high affinity of AVP to vasopressin receptor subtypes.

In general, the pharmacological approach is consistent and the data well presented, with no flaws in the experimental procedures.

Finally, concerning the form of the manuscript, it is rather synthetic; probably a more detailed and extended form will improved the manuscript.

Reviewer #3 (Remarks to the Author):

The manuscript “Structural basis for the activation and ligand recognition of the human oxytocin receptor” by Yann Waltenspühl and co-workers tackle important questions on OXTR function and structure, finally in association with OT and G-protein. The OT/OXTR interrelations are of high physiological importance and also of significance for diverse pathogenic conditions. Therefore, there was a strong need to clarify several details and general aspects concerning ligand binding and receptor regulation, which are now provided progressively. Even the resolution with 3.2Å is not very high, the structural part and additionally provided data are together a solid analyses of the targets, specifically the binding mode of OT is deeply analyzed, and related unique TM7 modifications are well investigated. This work also extends in a complementary way the previously from the authors provided crystal structure of OXTR in an inactivated state, bound with an antagonist. In addition, the comparison with e.g. known information from the V2 receptor makes the discussion comprehensive.

However, as known for sure a recent work of Justin G. Meyerowitz and colleagues published in March 2022 (Nature Structural & Molecular Biology) did already describe most of the findings reported here. Moreover, their cryo EM complex-structure resolution is higher with 2.9 Å and, therefore, allows also to explore the detailed role or binding mode of magnesium at OXTR. This weakens of course the novelty-attribute of the submitted manuscript (page 8, line 155), while the quality of the current work is high.

To my opinion the manuscript is a well written report in the claimed frame. Therefore, just few suggestions for potential improvement:

1. Structural and functional findings should be now compared also with the above mentioned new cryo-EM structure, and, particularly, potential differences must be extracted to highlighted putative specificities.

To add more structural-functional information beyond known insights would of course increase the impact of this study.

2. Page 6, line 94, figure 2a: This figure must be improved, e.g. the mentioned NPxxY motif is actually not visible, and the inserted windows are also very tiny. Figure 2 – which includes the most important visualized information - might be splitted into 2 figures, enabling the enlargement of each a-e parts, for clarity reasons.

To facilitate reading we have copied the reviewers' comments in italics and given our responses directly underneath in plain text.

Response to reviewers' comments:

We were of course very pleased by the favorable assessment of the reviewers, and we thank them for their insightful comments. Our response to the reviewers' comments and a detailed description of the changes we made in the manuscript are listed below. All changes are marked in yellow in the revised manuscript.

Reviewer #1:

Remarks to the Author:

The manuscript by Waltenspühl Y et al. describes the three-dimensional structure of the active human oxytocin (OT) receptor (OTR) in complex with its natural ligand oxytocin, a chimeric mini-Go/q/i protein, and the stabilizing antibody fragment ScFv16. The study enabled single-particle cryo-electron microscopy (cryo-EM) analysis of the OT:OTR:Go/q/i:ScFv16 complex at a resolution of 3.2 Å allowing to provide a structural basis for the activation and ligand recognition of this G protein-coupled receptor (GPCR) which plays a central role in the regulation of social behavior and reproduction.

This work is particularly important because it is complementary to the publication of the crystal structure of an antagonist-bound OTR by the same research team 2 years ago (the antagonist being the nonpeptide retosiban). Interestingly, the comparison between the inactive and active states of the receptor is quite useful to increase our knowledge of molecular determinants involved in its activation mechanism. In the future, this structural information will also help in the rational design and development of novel OT-based ligands for treating socio-behavioral pathologies.

The expertise of the authors in the scientific field of GPCRs and structural biology is internationally well renowned, the methods developed are state-of-the-art and the results presented in the article are convincing. In addition, the study is in total agreement with those describing the active structures of the OTR-related arginine-vasopressin (AVP) V2 receptor (V2R) in complex with AVP and its canonical Gs signaling protein, that were published last year.

Response: We thank the reviewer for this favorable comment.

I would re-organize the article for several reasons. First of all, I do not understand why only 2 principal figures are included in the main document. Second, some panels, for instance those in figure 2b and supplementary figure 7, are shown twice. Third, some close-up illustrations do not correspond to their corresponding squared dashed lines. Finally, some data illustrated in the figures are poorly

discussed. I think the format of Nature Communications Journal is quite flexible and because the present document is short, the main text may be lengthened.

Response: We thank the reviewer for his/her detailed suggestions; we have edited the figures accordingly. We agree that this change adds to the clarity and that there is no necessity to keep important information in the Supplement.

a) In the main text (page 3, second paragraph, line 32), while the authors claim that OT-based treatments of socio-behavioral deficiencies have yet failed, they may add a reference about the first nonpeptide OTR agonist that improves social interaction in a mouse model of autism. Indeed, LIT-001 can penetrate the blood-brain barrier and displays potential advantages of structural simplicity, low molecular weight, and drug-likeness as compared to OT (Frantz MC et al., J Med Chem, 2018, 61(19), 8670-92). However, indeed, this compound is not highly selective to OTR.

Response: We have mentioned this finding and added the corresponding reference to the manuscript.

b) In the main text, pages 4 and 5, while describing the OT binding pose in the receptor orthosteric binding site, the authors should acknowledge that their results are in accordance with previous data. AVP and OT are two chemically-related hormones (two cyclic nonapeptides with only 2 different residues at positions 3 and 8) and their targets constitute a high-homology receptor family sharing a common conserved binding site. At least, the authors should cite the first experimentally-validated model of AVP binding in V1aR published in 1995 (Mouillac B et al., J Biol Chem, 270(43), 25771-77), since the identification of receptor residues involved in the binding of both hormones in the two studies is consistent. Hopefully, more recent papers (Bous J et al., Sci Adv, 2021, 7, eabg5628; Zhou F et al., Cell Res, 2021, 31, 929-31; Wang L et al., Cell Res, 2021, 31, 932-34) where cryo-EM active structures of AVP-bound V2R in complex with Gs protein have been determined, are currently included in the reference list.

Response: We have included the references into the manuscript and would like to point out we had already included a discussion to the recently published V2R structures.

c) A paper by Meyerowitz JG et al., very recently accepted for publication in Nature Structural Molecular Biology (it can be seen in the latest research articles, 2022, March 3rd, on the web page of this journal), should be added as a reference. In this article, the cryo-EM structure of the OT-bound OTR receptor in complex with a mini-Gq/i (at a resolution of 2.9 Å) is described. It is in complete agreement with that described in the present paper by Waltenspühl et al. Interestingly, the two papers

are complementary. This study may be cited on page 4, lines 58-59. The authors should indicate that their study, in agreement with that by Meyerowitz and colleagues, enabled unprecedented insights into OTR activation by OT and into OTR-G protein interaction

Response: When we initially submitted the manuscript, we did not know of the existence of the work of Meyerowitz *et al.*, which was only published online subsequently. We have now, nonetheless, added a sentence to the manuscript referring to the work of G. Meyerowitz *et al.*

Reviewer #2:

The authors present a crystal structure of the Oxytocin receptor (OXTR), at a resolution of 3.2 Å, which, is not particularly high in the context of the most recent solved membrane protein structures. Recently, the same authors reported another OXTR structure (Crystal structure of the human oxytocin receptor. Waltenspühl Y, Schöppe J, Ehrenmann J, Kummer L, Plückthun A. Sci Adv. 2020). The previous structure was solved by employing a small non peptidic antagonist (retosiban), which enabled the stabilization of a presumably inactive receptor conformation. Not surprisingly, the orthosteric binding site defined in the present paper overlaps with the one already described in the previous structure and with previous models obtained by homology modelling and site directed mutagenesis. Shifts of extracellular and intracellular helix tips and Class-A specific microswitches (in the CWxP, PIF, NPxxY, DRY) from inactive to active states are reported in extended data fig.5, with no major breakthrough. Several residues described here as relevant for binding have been already subjected to site-directed mutagenesis, binding and activation studies, and it would have been appropriate to cite and discuss this literature (which the authors could easily find and access).

Response: We partially agree with the reviewer in that of course the increase in knowledge is more incremental today than it was in 2007 when the first non-opsin structures were published. The fact that the shifts of some of the microswitches and helical tips upon activation are similar as in some other receptors is perhaps in hindsight not surprising, but is not at all uniform across receptors, and therefore this gain in information is still highly valuable. We would like to reiterate that a high-resolution structure is currently still the only method to validate previously proposed structural knowledge gained from homology modelling, docking and site directed mutagenesis. We would like to also emphasize that studies prior to the structural work had already been mentioned in our manuscript and were reflected in our reference list. Importantly however, those methods, even in combination, unfortunately cannot properly discriminate between allosteric and direct effects by themselves. For example, Chini *et al.* 1996 predicted two aromatic residues to be located at the

bottom of the OT binding pocket. With our structural knowledge, we can now clearly show that this is not the case. The described loss of activity when either of those residues are mutated, although real, is hence clearly due to allosteric effects measured.

As an example, concerning the discussion of “Early functional studies on OT derivatives” (line 98), the authors should refer to the extensive work reported by Berde and Boissonnas (In Berde, B. (ed.), 1968, Handbook of Experimental Pharmacology. Neurohypophysial Hormones and Similar Polypeptides. Springer-Verlag) in which substitutions of all residues of OT and AVP with a number of L- and D- aminoacids (including O-MeTyr at position 2) were reported; this work remains a milestone in the field and constitutes the ground for the following years of OT/AVP peptide design and synthesis (see for example Hruby and Chow, M.S. (1990) Annu. Rev. Pharmacol. Toxicol; Hruby and Smith (1987) In Smith, C.W. (ed.), The Peptides. Chemistry, Biology and Medicine of Neurohypophysial Hormones and Their Analogs. Academic Press). In particular, please note that alkylation of residue 2 was already reported in 1968 by Berde, B. and Boissonnas, R.A to confer antagonistic properties to OT/AVP ligands and extensively exploited to design potent and selective OXTR antagonists (Manning and Sawyer, J Rec Res 1993). I suggest that integrating their actual data with the data already present in the extensive OT/AVP literature could help to identify new clues for the development of OT-based therapeutics, as stated in the conclusion, improving the interest of the manuscript. This also apply to the discussion of the OTR/V2 structure, as many ligands display selective agonist/antagonists (biased agonists profiles at these receptors: could the authors provide new clues for the design of more selective/potents analogs?)

Response: We thank the reviewer for this comment, and we have now included the citation of Berde and Boissonnas in our manuscript. We would like to mention that we have solved the structure in complex with only the natural agonist oxytocin. A thorough and complete structure-function relationship of all OTR ligands would not only require further structural studies and/or modelling but also an extended physiological characterization of all these ligands and is thus in our opinion outside of the scope of this work. We would further like to point out that we do not claim that we have the unique answer to all these questions but that our structure is the foundation to now tackle this work.

Lines 132-133 The sentence “in contrast, the binding conformation of AVp in V2R is compatible with OTR binding” is not clear to me. Please explain

Response: We changed this sentence to *“In contrast, the bound conformation of AVP in V2R is compatible with binding to OTR.”*

For further clarification we have extended our discussion and would like to draw the reviewer's attention to the modified and new sentences in our manuscript:

"Therefore, the positioning of helix 1 might explain why the OTR is not selective between OT and AVP, but OT binding is specific to OTR and it does not bind to V₂R. In V₂R, helix 1 is packed in a more compact manner in the helix bundle compared to the OTR so it would clash with Gly₉ if OT were to be expected to adopt the identical binding conformation as observed in complex with OTR."

We illustrate this in the figure below, but since this is somewhat speculative and relies a lot on the structure of V₂R, which is not our work, we do not wish to include this figure into the manuscript.

Lines 135-139: A direct interaction between F103 in the OTR and OT has never been proposed, to my knowledge: an interaction was however proposed at D and Y residues present at that position in V₂ and V_{1a} receptors to explain the high affinity of AVP to vasopressin receptor subtypes.

Response: We like to point out that this particular residue, F103 or D103, in fact packs on top of helix 2, pointing away from the binding pocket, thereby likely contributing to helix 2 stability. The similar orientation is also observed in the recently published structure of V₂R (see figure attached).

The effects measured during mutational studies could, therefore, also be caused by allosteric effects, i.e., through the destabilization of the receptor. Further high-resolution structures of the V_{1A}R or V_{1B}R will have to be determined to study this further. This particular residue thereby further underlines why mutational studies are not sufficient to map a binding site but structural knowledge for verification is in every case required.

In general, the pharmacological approach is consistent and the data well presented, with no flaws in the experimental procedures.

Response: We thank the reviewer for this comment.

Finally, concerning the form of the manuscript, it is rather synthetic; probably a more detailed and extended form will improved the manuscript.

Response: We have extended the manuscript to better suit the format.

Reviewer #3:

Remarks to the Author:

The manuscript "Structural basis for the activation and ligand recognition of the human oxytocin receptor" by Yann Waltenspühl and co-workers tackle important questions on OXTR function and structure, finally in association with OT and G-protein. The OT/OXTR interrelations are of high physiological importance and also of significance for diverse pathogenic conditions. Therefore, there was a strong need to clarify several details and general aspects concerning ligand binding and receptor regulation, which are now provided progressively. Even the resolution with 3.2Å is not very high, the structural part and additionally provided data are together a solid analyses of the targets, specifically the binding mode of OT is deeply analyzed, and related unique TM7 modifications are well investigated. This work also extends in a complementary way the previously from the authors provided crystal structure of OXTR in an inactivated state, bound with an antagonist. In addition, the comparison with e.g. known information from the V2 receptor makes the discussion comprehensive.

Response: We thank the reviewer for this favorable comment.

However, as known for sure a recent work of Justin G. Meyerowitz and colleagues published in March 2022 (Nature Structural & Molecular Biology) did already describe most of the findings reported here. Moreover, their cryo EM complex-structure resolution is higher with 2.9 Å and, therefore, allows also to explore the detailed role or binding mode of magnesium at OXTR. This weakens of course the novelty-attribute of the submitted manuscript (page 8, line 155), while the quality of the current work is high.

Response: We would like to point out that, while our findings are in some parts parallel to the observations of Meyerowitz (published after the present work had been submitted), there are still important differences. For example, i) we have included a more detailed analysis of the reorientation of helix 7, ii) we have solved the structure with a different G protein chimera and iii) while we were also initially tempted to model the Mg²⁺ into our structure, we have refrained from doing so as the proposed model (like the one proposed by G. Meyerowitz *et al.*) would describe an indirect coordination of Mg²⁺ to the OTR through water molecules. In our opinion, such a hypothesis is highly speculative and normally not found in Mg²⁺-requiring proteins

To my opinion the manuscript is a well written report in the claimed frame.

Response: We thank the reviewer for this favorable comment.

Therefore, just few suggestions for potential improvement:

1. Structural and functional findings should be now compared also with the above mentioned new cryo-EM structure, and, particularly, potential differences must be extracted to highlighted putative specificities.

To add more structural-functional information beyond known insights would of course increase the impact of this study.

Response: We have added a sentence referring to the recently published structure but have refrained from a detailed comparison of the two structures, as there are several differences in the complexes used, as explained above. We would also like point the reviewer's attention to the fact that the structures were probably simultaneously submitted, as our bioRxiv upload (<https://doi.org/10.1101/2022.02.21.481286>) is dated before the appearance of the work of G. Meyerowitz *et al.*.

2. Page 6, line 94, figure 2a: This figure must be improved, e.g. the mentioned NPxxY motif is actually not visible, and the inserted windows are also very tiny. Figure 2 – which includes the most important visualized information - might be splitted into 2 figures, enabling the enlargement of each a-e parts, for clarity reasons.

Response: We agree that our initial manuscript figures were not ideally formatted. We have, therefore, extended our figures accordingly.